# Mixing Oil with Water: Framing and Theorizing in Management Research Informed by Design Science

A. Georges L. Romme [1],* and Dimo Dimov [2]

1 Department of Industrial Engineering & Innovation Sciences, Eindhoven University of Technology, P.O. Box 513, 5600 MB Eindhoven, The Netherlands

2 School of Management, University of Bath, Claverton Down, Bath BA2 7AY, UK; d.p.dimov@bath.ac.uk

* Correspondence: a.g.l.romme@tue.nl

**Abstract:** Design science (DS) approaches have been emerging in engineering, management and other disciplines operating at the interface between design research and the natural or social sciences. Research informed by DS is challenging because it involves "mixing oil with water", using a famous phrase of Herbert Simon. A key challenge here is the dual role of theory: one can develop a "theory of" any empirical phenomenon to explain its characteristics and outcomes, or alternatively, develop a "theory for" generating this phenomenon, focused on solving problems and enlarging possibilities. To clearly distinguish these two perspectives, we talk about *theorizing* in relation to theory-of and *framing* related to theory-for. A state-of-the-art review of how DS is applied by management researchers results in two main findings. First, explicit (re)framing efforts appear to be highly instrumental in challenging a given theoretical paradigm and thereby reduce the risk of being constrained to it; these findings confirm the generative nature of design activity. Moreover, many studies reviewed draw on knowledge formats that synthesize descriptive-explanatory and prescriptive-normative knowledge. Our main findings are subsequently integrated into a DS methodology, which may especially be of interest to design-oriented disciplines that tend to adopt a rather intuitive (undefined) notion of theory.

**Keywords:** design methodology; theory; design science; design research; validation; management studies; framing; generativity; research methodology

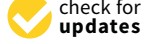



## 1. Introduction

In the history of ideas in Western philosophy, there has been a persistent tension between being and becoming. These two views draw on different philosophical approaches and interpretations of human experience, which come to a head in our understanding of human artifacts as either natural or artificial objects [1]. In their natural sense, artifacts are taken for granted and thus serve as a building block for describing and explaining the world around us. In their artificial sense, they can be conceived as the outcome of human agency, designed for specific purposes in a world in which they did not yet exist. The quest for knowledge continues to be energized by this tension between the actual and the possible, between the certainty of facts and the uncertainty of possibilities.

Simon's seminal work [1] has inspired the rise of design science (DS) approaches focused on the exploration of possibilities against a background of natural and social sciences building on facts. DS approaches have thus been emerging in engineering [2], information systems [3,4], organization and management studies [5,6] and entrepreneurship research [7]. As a result, a growing number of scholars have been applying DS methods (e.g., [8–12]). Simon's initial conception of DS was severely criticized [13–15] and, therefore, DS has further developed in highly different ways [2–6,14–17]—which suggests Simon's initial idea of DS primarily operates as a metaphor that can be used and molded in many directions.

Research informed by DS raises major challenges, especially in disciplines in which scholarly status and legitimacy have been tied to the rigor of their knowledge base. Accordingly, the methodology of the natural and social sciences almost completely drove out the (pragmatist) design approach from engineering as well as business schools in the 1950s and 1960s, primarily because scholars and deans hankered after academic respectability, with the natural and social sciences as their main role models [17–20].

Subsequently, design research started flourishing in engineering and related disciplines despite this initial setback [2–4], but the field of management continues to suffer from its overcompensation for past scars of academic insecurity. That is, a vast majority of management scholars avoid doing problem-solving design work and focus on developing explanatory-descriptive theories [21–23]. In the 1960s, Simon [24] already anticipated that mainstream science would drive out the design approach (from management schools). That is, combining design and science is very much like "mixing oil with water": it is rather easy to describe the intended outcome but rather difficult to produce it. Moreover, the task of mixing oil and water is not completed when the two components are mixed; left to themselves, they will start separating again [18,24]. The current intellectual stasis of management scholarship and its growing detachment from business practice [21,22] underline how difficult it is to mix science and design.

A major challenge in DS appears to arise from the *dual* role of theory. That is, theories can shape the interplay between the (empirical) world and its conceptual representation in two ways: one can focus on developing a *theory of* the world by means of constructs and models describing and explaining it, or alternatively, develop a *theory for* the world and use it as a gateway for discovering any objects behind it or creating entirely new objects [25,26]. This theory of-for distinction appears to divide scholars in the field of management—to the extent that they tend to limit the use of theory to one or the other. That is, theorizing or theory is conceived either as "a way of imposing conceptual order on the empirical complexity of the phenomenal world" [27] (p. 407) (see also: [28–30]) or alternatively, as an act of creative framing and discovery that serves to deeply understand and possibly better manage, the phenomenon at hand [31–33].

To clearly distinguish the two, we refer in the remainder of this paper to *theorizing* when developing theories *of* management practices and to *framing* in case of theories *for* (generating) these practices. The next section sets the stage for our literature review and synthesis by exploring the theory of/for dispute in management research and assessing the discourse on theory development in the DS literature. Subsequently, we review how DS approaches have been used by management scholars, with a particular focus on the interplay between framing and theorizing.

Our paper serves to reflect on the body of work that has been arising from various conceptual pieces on DS [5,6,14], all inspired by Simon [1]. First, we find that explicit (re)framing efforts appear to be highly instrumental in challenging a given theoretical paradigm and thereby reduce the risk of being constrained to it. Second, in developing theory in the service of practice, DS work often draws on a knowledge format that connects and integrates descriptive-explanatory and prescriptive-normative knowledge. Finally, we synthesize the various DS applications in a methodology that incorporates both framing and theorizing.

## 2. Background

To set the stage, Section 2.1 explores the theory of/for dispute in management studies. Subsequently, we review the discourse on theory development in the DS literature (in Section 2.2). The notion of "design science" used in this paper is distinct from the so-called "science of design" literature that largely focuses on developing a "design theory", that is, a deep understanding of the ontology and knowledge structure of design activity [16,34]. By contrast, the DS discourse focuses on the interface and interaction between design and (explanatory) science. In this paper, we, therefore, seek to "practice what you preach" [35,36] by engaging in a descriptive and explanatory analysis of the state-of-the-art of DS in man-

agement studies, which subsequently informs the design of a synthesized framework. As such, the methodology of this study is reflexive [36] in the sense that it operates at the interface of design and science. Notably, we do not discuss the pragmatist underpinnings of DS in this paper, which are extensively described and discussed elsewhere [37,38].

### 2.1. Developing Theories of and for Practice

Fundamentally, theory—in its Greek origin "theoria"—implies observation, beholding. Observation involves a focal subject (i.e., the observer), some external object (i.e., what is observed), and a relationship between the two (i.e., the purposeful act of observation). By taking the act of observation and thus theory for granted, scholars may overlook the generative role of the relationship between subject and object. In this respect, human observation and theorizing are embedded in broader, purposeful human activities mediated by conceptual/representational tools [1]. Accordingly, a theory is a symbolic representation of something external, that is, a conceptual tool or map for engaging with the world. Given that "a map is not the territory it represents" [25] (p. 58), scholars need to be aware of the distinction between objects and their conceptual representations. What gets on the map is "difference", which is not an inherent feature of the territory, but primarily reflects the stance of the observer; the difference is an abstract concept involving schemes of categorization that helps one see things as similar or different [39], for example, "employees" versus "managers" or "income" versus "costs".

Theories can shape the interplay between the (empirical) world and its conceptual representation in two ways. One can focus on developing a theory of the world by means of constructs and models describing and explaining it, or alternatively, develop a theory for the world and use it as a gateway for discovering any objects behind it or creating entirely new objects [1,26]. This theory of/for distinction may divide researchers to the extent that they limit the use of theory to one or the other (e.g., [27,32]).

### 2.2. Theory Development in Design Science

Simon [1] argued that "science" develops theoretical knowledge about what already is ("facts"), whereas "design" is about using knowledge to create what should be, things that do not yet exist ("artifacts"). Design, as the activity of changing existing situations into desired ones, therefore appears to be a core competence of all professional activities, including engineering and management: that is, "everyone designs who devises courses of action aimed at changing existing situations into preferred ones. The intellectual activity that produces material artifacts is no different fundamentally from the one that prescribes remedies for a sick patient or the one that devises a new sales plan for a company" [1] (p. 111). The "mixing oil with water" challenge of combining design and science has been acknowledged in the management and entrepreneurship literature [5,7] as well as in adjacent fields such as engineering design [2], information systems [3] and philosophy of science [40].

In the engineering design literature, the notion of theory has especially served to develop a "theory of design", that is, a deep understanding of the ontology and knowledge structure of design activity [16,34]. For example, generativity has been identified as a core element of design reasoning, and the so-called splitting condition was found to be an important prerequisite of generativity [34,41]. In general, the literature on design theory emphasizes the complementarity and synergy between theorizing about and for design activity; that is, they are conceived as inseparable [13,42]. Consequently, this literature has not explicitly addressed the "mixing oil with water" challenge that appears to prevail in other fields, such as management and organization studies, in which descriptive and explanatory science continues to be highly antagonistic to creative and generative design.

Only the DS framework proposed by Holmström, Ketokivi and Hameri [43] explicitly addressed theory development. They argue that the development of an artifact is what distinguishes DS from other research approaches, such as action research. DS thus serves to develop "a means to an end" [43] (p. 67), that is, an artifact is designed to solve a problem,

implying the analysis of the present state, the desired state, and the design of solutions that may help move from the present to the desired state. Accordingly, Holmström and coauthors distinguish four phases: solution incubation, solution refinement, explanation I—substantive theory, and explanation II—formal theory. Solution incubation and refinement consist of framing the business problem, developing a preliminary solution design (detailed enough to be evaluated and tested) and iteratively refining this solution by exposing it to testing. This phase involves synthesizing inputs from multiple disciplines and, as such, requires abductive framing in spotting the commonalities across different perspectives. Solution refinement usually completes the process for practitioners, who are likely to stop when a satisfactory solution has been developed [43].

Accordingly, Holmström and coauthors argue that the theory development phases are predominantly the domain of scholars who seek to generalize the findings and make a scientific contribution [43]. In the substantive theory phase, the solution design is evaluated from a theoretical point of view to produce and advance a relevant theory. Such theory is context-dependent and has a limited scope of applicability. Since it is valid only in some type of context, arguments with respect to the contextual boundaries are an important part of the substantive theory. From a means–ends perspective, the objective of this phase is to generalize the research findings in a theoretical sense through systematically implementing the solution in contexts in which the means–ends proposition is relevant [43]. In developing formal theory, DS research seeks to develop formal theoretical propositions that are not limited to the empirical context of the solution design initially developed: "Empirical examples can be used to illustrate the theory, but the theory itself can in a sense stand on its own feet" [43] (p. 76). Formal theories often develop from substantive theories, but they seek broader generalizability in terms of theoretical abstraction as well as statistical generalization. The means–ends propositions that constitute the heart of a formal theory are thus theoretical abstractions that are embedded in the logic of the theory itself. For example, a means–ends proposition in transaction cost theory is as follows: in order to minimize transaction costs (cf. end) in conditions of uncertainty, frequent transactions and high asset specificity, one should internalize the transaction (cf. means) [43].

Overall, the notion of theory and theorizing is underdeveloped in the discourse on combining generative design and explanatory science. Here, the field of management research appears to provide a highly interesting setting to study the conceptualization and application of theory because "mixing oil with water" is a major challenge in this field [22,44].

## 3. Review Scope and Approach

This section explores in more depth what can be learned from how DS-based work in the field of management engages in framing and theorizing. We adopted an integrative review approach that serves to assess and synthesize a set of publications in an integrated manner, such that new perspectives and frameworks can be generated [45]. For this review, we selected a sample of DS studies published since 2002–2003, when Hatchuel [2] and Romme [5] published their conceptual DS frameworks, later followed by Van Aken [6]. The studies were selected using three criteria: the publication needed to (a) be explicitly positioned as drawing on or being informed by DS, that is, it is explicitly labeled as "design science" or similar terms like "science-based design" or "design-based research"; (b) go beyond the assignment of merely creating a practical solution/artifact, by also engaging in some form of theory development; and (c) focus on objects commonly studied by management scholars. The second criterion implies that entirely conceptual pieces on DS were excluded, while several papers that review and synthesize the literature on a particular problem in terms of design principles (e.g., [46,47]) were included. The third criterion implies we selected studies addressing topics like organizational development, knowledge exchange, management innovation, corporate leadership, business models, digital marketing, and so forth.

This selection process resulted in 28 initial sources [8–12,46–68]. In several instances, the selected publication appeared to be part of a larger research program, implying we also reviewed various (preceding or subsequent) publications connected to the initial source. This resulted in another 20 sources reviewed in the following section.

## 4. Main Observations and Patterns in How DS Is Used

In view of the tensions and complementarity between theorizing and framing outlined earlier, our review in this section focuses on how authors engage in framing and theorizing. We describe two key findings. First, DS appears to be instrumental in reducing the risk of being constrained to an extant theoretical paradigm. Second, to bridge the design and science modes, DS-based work often draws on knowledge formats that effectively connect and integrate retrospective science and prospective design.

### 4.1. Deliberate Framing of the Problem Serves to Avoid Being Captured

Many DS-based studies serve to challenge prevailing theories (e.g., [52,53,56,59]). For example, Elmquist and Le Masson's [53] study starts from the observation that prevailing quality–cost–time (QCT) frameworks cannot be applied to innovation projects for which specifications are not known at the beginning. These authors, therefore, framed their study around the question of whether discontinuous innovation projects can be evaluated in terms of how they contribute to building the firm's innovative capabilities. Based on a review and synthesis of the literature as well as a case study at RATP, the largest French public transport provider, Elmquist and Le Masson, created a new framework for evaluating projects as "interdependent" activities, using evaluation criteria related to the innovation capabilities of the firm. This study resulted in the novel insight that even a project that appears to have failed (according to QCT criteria) can still generate a substantial amount of other value for the company [53]. Later work extended some of these initial insights [69,70].

Romme and Endenburg [59] challenged prevailing theories and practices of organizational democracy and corporate governance. The initial framing of this study was based on the governance dilemmas and tensions which Endenburg experienced as managing director of a company. Therefore, he sought to develop a corporate governance approach that would promote a genuine dialog between management and employees instead of frequently producing conflict. This informed the development of a novel set of design principles for organizational democracy and governance, drawing on the notion of circularity from cybernetics. These circular design principles served to develop and implement a circular structure in the company led by Endenburg (constituting an alpha test in DS terms); this new structure was effective in bringing people, information and knowledge together to solve problems and make policy (see also [71]). In addition to several other applications of the circular organization design [58,59], this design later inspired the development of similar organizational designs, the most prominent one being Holacracy [72], which for instance, is applied in Amazon.

Other work illustrates the capability of DS scholarship to discover and explore entirely new theoretical territory [48,63,67]. For example, Andriessen [48] set out to develop and test a tool at KPMG for reporting intellectual capital (IC) in the absence of a theoretical body of knowledge on IC. He framed this tool as an Organizational Development (OD) intervention aimed at influencing the individual and collective sense-making of managers. The key idea was that a change in managerial sense-making regarding IC would result in the better and more sustainable management of intangible resources. Andriessen developed a list of requirements and inferred design propositions from adjacent fields (i.e., core competence theory and valuation theory) to create a first version of the tool. The tool was then tested in six companies in various industries and with various sizes, allowing for both alpha and beta testing; that is, the lead designer (i.e., Andriessen) was not involved in two of these tests. These tests served to identify the conditions under which the tool works (i.e., for long-term strategy development in knowledge-intensive, medium-sized companies) or does

not work. The tests also served to uncover the generative mechanisms (e.g., appreciative framing) that made some interventions effective and others not. Andriessen codified these findings in various propositions that describe the class of problems and the class of contexts for/in which IC reporting appears to be effective; and several propositions that depict the interventions and generative mechanisms of IC reporting; for instance: "The IC reporting tool is successful in creating energy with members of an organization because it uses a mechanism of appreciative framing. This provides a new, more positive view of the company and helps develop a common language that can explain the company's success" [48] (p. 102).

Overall, these studies demonstrate that DS supports generative reasoning [34] and is therefore likely to result in ideas and propositions that challenge a given theoretical paradigm or create an entirely new one. Deliberate framing efforts appear to drive these paradigm shifts. DS can thus serve to reduce the risk of being constrained to an extant theoretical paradigm by challenging the latter from the perspective of new frames. This key finding is in line with the key role of generativity in design reasoning [34,41].

*4.2. Connecting Retrospective and Prospective Knowledge*

To connect design and science, and in particular framing and theorizing activity, many DS-based studies develop knowledge that combines descriptive-explanatory and prescriptive-normative elements [5,61,67,68,73]. In particular, so-called design propositions or principles appear to be instrumental in connecting the body of scientific evidence to prospective design: as the "real helps" of managerial thought and action [33] (p. 130), they link the descriptive and explanatory nature of (most) scholarly efforts to the normative and situated nature of work by practitioners [51,59–61,67].

Denyer and coauthors [73] draw on the research synthesis literature to propose the CIMO format for design principles. In this format, a design principle depicts the *context* in which an *intervention* that activates a particular *mechanism* is likely to result in a specific *outcome*. For the entrepreneurship field, Van Burg and Romme [65] developed a similar context–mechanism–outcome (CMO) format, arguing that contextual conditions and social mechanisms together inform a (typically rather broad) domain of actions within which entrepreneurs and their stakeholders can operate in order to accomplish their intended outcomes. These knowledge formats have been widely applied [8,9,46,48,49,60,64,67].

For example, Hodgkinson and Healey [46] draw on various literature works to distill a set of design propositions that would inform the design of scenario planning interventions, especially in the area of facilitation and team composition. One of their propositions is: "When working with an informationally diverse scenario team, to reduce inter-subgroup bias and facilitate the elaborative processing required for effective scenario construction and analysis, stimulate superordinate recategorization by emphasizing the shared fate of the scenario team and establishing common goals" [46] (p. 444).

In another study, a solution for knowledge management was developed for a cluster of telecom companies residing on a French campus [9]. This study served to develop a set of CIMO-formatted design propositions based on a literature review and interviews with (future) users of the solution. The first proposition is: "In a multi-actor cluster with a broad scope of technologies (C), an interactive map of competencies (I) will serve to foster knowledge creation through R&D collaboration (O) by reinforcing the four potential mediators of knowledge creation: opportunity, anticipation ability, motivation and combinative capability (M)" [9] (p. 271). The latter mechanisms were subsequently fleshed out in several other propositions, for instance, regarding the opportunity mechanism: "In a multi-actor cluster with a broad scope of technologies (C), an interactive map of competencies (I) provides relevant information that enhances opportunities (M) for finding a good partner for R&D collaboration (O). To trigger the opportunity mechanism, a competency is defined as an action that mobilizes technical, scientific and managerial resources (including knowledge) to produce deliverables that are likely to create value in business activity" [9] (p. 272).

A final example is the study by Sagath et al. [60], who developed a set of design principles for new business incubation in the space sector by synthesizing previous studies and combining these with practitioners" experiences documented in three incubators in the European space sector. All these examples demonstrate that a design principle tends to be part of a bundle of (complementary) principles. Each principle provides themes on which managers and scholars can write their own variations [5,59], which is also evident from words like "stimulate", "foster", and "enhance" in the previous examples. In more formal terms, design principles appear to incorporate an ambiguity operator in predicate logic: in context C, doing something like action A will help activate mechanism M, which is likely to realize something like O [74].

The DS literature thus suggests that a knowledge format connecting prospection and retrospection is instrumental in promoting theory development at the interface between design and science. The next subsection serves to further codify these review results.

*4.3. CAMO-Formatted Knowledge: Codifying Design Science Practice*

The previous section suggests that a knowledge vehicle that synthesizes prospective and retrospective perspectives as well as cross-fertilizes academic and managerial work can be instrumental for promoting dialog across the design and science modes. We codify this insight here in terms of a context–agency–mechanism–outcome (CAMO) format. This format is grounded in the literature on research synthesis [65] and is adapted from the CIMO-format proposed by Denyer et al. [73]. The key difference with CIMO is that the latter leaves the *agent* dimension somewhat unspecified by merely describing the intervention/action (and not who the agent is behind this intervention); the main difference between CIMO and CAMO, therefore, is that the intervention dimension is replaced by the agency dimension.

The basic rationale of CAMO is that any form of knowledge, to be useful and actionable as a conceptual map, needs interfaces that facilitate its deployment and use in a practical situation. As such, it requires specifications of its boundary conditions, the actions it enables or informs, the outcomes it can produce, and the mechanisms via which such outcomes are generated. Such specifications enable problem-solving via systematic consideration of whether the situation, actions, outcomes, or mechanisms are appropriate and valid. In Aristotelian terms, these specifications can be seen as a constellation of the four causes of why things come into being:

- *final* cause—the outcome, for the sake of which it comes into being;
- *efficient* cause—the agency that initiates the change;
- *formal* cause—the mechanism that operates as the shaping force; and
- *material* cause—the context providing the immanent elements [75].

In line with the pragmatist underpinnings of DS, the CAMO format avoids specific ontological assumptions that assign meanings to (otherwise merely) nominal categories [76,77] and thereby goes beyond the realism-constructivism debate. Notably, we call CAMO a "format" because of its generic nature: that is, the CAMO format can be used to, for example, theoretically explain empirical phenomena, articulate design principles that inform the creation of artifacts, and codify research findings. In terms of the distinction between framing and theorizing, the A and O elements tend to be central to the framing act (i.e., an agent seeks to solve a particular problem, that is, create an intended outcome), whereas full-fledged theorizing also needs to address the C and M elements, that is, the boundary conditions and generative mechanisms. However, the generative nature of design work implies that framing activity can and should not be constrained; it may therefore also involve (initial) explorations of the C and/or M elements [53,59,67]. The key idea here is that the CAMO format appears to be highly instrumental in connecting the prospective nature of design (including framing) to the descriptive and explanatory nature of science (including theorizing). Later in this section, we will explore the role of CAMO in developing substantive and formal theories. In the remainder of this subsection, we define each element of CAMO.

*Context.* The context of knowledge application pertains to the (boundary) conditions of a key structural feature of interest, such as supervisor-employee or investor-entrepreneur relationships. These conditions provide the specific "materials" from which a structural feature is made. Contextual conditions can enable or constrain the choices and behaviors of actors in and around organizations [28]. While actors in and around organizations typically have discretion in making choices, a specific context tends to direct and/or restrict these choices. In general, the key role of contextual conditions in CAMO is grounded in institutional and structurationist perspectives [78,79].

*Agency.* Knowledge application requires certain instrumentality or agency, that is, a sense of who can initiate which actions and thus operate as an efficient cause. Agency is the capacity of an actor to act in a given context. Emirbayer and Mische [80] provided a fine-grained definition of agency as "a temporally embedded process of social engagement, informed by the past (in its "iterational" or habitual aspect), but also oriented toward the future (as a "projective" capacity to imagine alternative possibilities) and toward the present (as a "practical-evaluative" capacity to contextualize past habits and future projects within the contingencies of the moment)" (p. 962). The agency dimension is grounded in action research [81], practice theory [82] and related studies. While the notion of agency has been disputed in terms of its embeddedness and relationship to structure/context [30,83], the DS perspective merely implies this notion needs to incorporate a retrospective as well as prospective orientation—as in the definition of Emirbayer and Mische.

*Mechanism.* Knowledge application comes with an (often implicit) "theory of change", involving expectations of the mechanisms that will be engaged, serving as a formal cause or blueprint. A mechanism is a causal driver that gives rise to a particular type of outcome, and an observed outcome can thus be explained by referring to the mechanism by which this outcome is regularly brought about [84]. As such, social or generative mechanisms [73,84] constitute a pivotal notion in research synthesis and theory development because a coherent body of knowledge requires some form of agreement on which mechanisms and associated actions generate certain outcome patterns in particular contexts [65]. Examples are causal drivers such as "clarity of goals and purpose", "situated hands-on practice", "stakeholder participation", "information sharing" and "use of incentives" [52,54,62]. Often, these mechanisms cannot be observed directly, implying that conceptual and analytical work is required to identify them and explain why, for example, outcome A occurs in context B. The mechanism dimension has its roots in the sociological literature on mechanisms, which draws on a pragmatist notion [76,84] in line with the DS perspective.

*Outcome.* Knowledge applications serve purposes or ends, which operate as a final cause to any managerial effort. Management scholars and practitioners share a strong interest in (un)intended outcomes [73], that is, empirically observable (patterns of) results—such as shareholder value, productivity, corporate failure, intraorganizational power distance, employee satisfaction, or employee engagement. A key assumption in CAMO is that there are no straightforward mechanisms explaining this type of outcome. Instead, CAMO goes beyond simple (e.g., agency-based) explanations of outcome regularities because many different—possibly unobserved—contextual conditions and social mechanisms may affect the outcome [85]. CAMO thus refers to which kind of agency, in a given Context, is likely to activate a particular (set of) mechanism(s), which together generate a particular Outcome pattern.

In sum, the CAMO format enables the integration of descriptive-explanatory and prescriptive-normative knowledge. Table 1 defines the four dimensions of CAMO and demonstrates how the component relationships (e.g., between M and O) are descriptive and explanatory in nature, which serves to firmly connect any DS-based study to the evidence available in the literature (e.g., [9,48,49,62,63]). However, the synthesized CAMO proposition is not only descriptive/explanatory but also prescriptive-normative in nature (see Table 1). It thus invites subsequent attempts by scholars to replicate and extend the proposition as well as informs practical work in similar contexts (e.g., by academic-practitioner teams), as illustrated in Section 4.2.

**Table 1.** The context–agency–mechanism–outcome (CAMO) format, defined.

|  | Context | Agency | Mechanism | Outcome |
|---|---|---|---|---|
| *Definitions* | Conditions that can enable or constrain (e.g., managerial) behavior and choices (cf., Aristotle's material cause). While many actors have discretion in making choices, these conditions tend to direct and/or restrict these choices. | The capacity of a specific (group of) actor(s) to act in a given context (cf., Aristotle's efficient cause). Agency, here, therefore, refers to both the actor(s) and its/their actions. | Driver, or Aristotle's formal cause, gives rise to a particular kind of outcome. Mechanisms can often not be directly observed; further conceptual and analytical work is then required to identify them. | The intended or unintended results of the combined agency–context–mechanism; the agent's intended result reflects Aristotle's final cause. |
| *CAMO Synthesis* | General structure of a CAMO proposition: *If in context C agency A activates mechanism M, this is likely to generate outcome pattern O.* This proposition thus explains the O from a specific CAM combination, with M being the key causal driver, A the activator of this cause, and C the boundary condition. The various component-relations of any CAMO proposition (e.g., A–M or M–O) are descriptive and/or explanatory in nature. However, the synthesized (set of) CAMO proposition(s) is descriptive-explanatory *as well as* prescriptive-normative in nature. Thus, the CAMO proposition above can be rewritten as a design principle as follows: *To generate outcome pattern O in context C, do something like A to activate mechanism M.* | | | |

### 4.4. Benefits of CAMO-Formatted Knowledge

There are several ways in which DS-based work can benefit from the CAMO-format bridging prospective and retrospective knowledge. We infer the following three benefits from the literature reviewed.

*CAMO informs substantive theory & AMO drives formal theory.* The application of CAMO or a similar framework serves to synthesize a fragmented body of theoretical knowledge and thereby motivate further theoretical work [12,47,52,54,61,62,64,67]. For instance, in exploring how (formerly self-contained) corporate ventures can be effectively integrated into established corporations, Van Burg et al. [64] synthesized the highly fragmented literature on strategic fit, transition timing, performance management and related topics in six design principles, including for instance: "Prepare venture transition by composing a dedicated transition team, conducting a readiness and capability assessment, and developing a transition plan, serving to enhance the integration process and avoid integration problems afterward ( . . . )." "The corporate venturing unit and the receiving business unit should jointly assess the transition timing. The best moment for transition is after the corporate venture has achieved the first sales and when support and assets of the established business become necessary to enable further growth" [64] (p. 467). The main contribution of the study by Van Burg and coauthors is the development of a process theory of the corporate venture transition process in each of its phases (pre-transition, transition, and post-transition). This process theory is formulated as a set of six design principles, which not only provides an instrumental framework for practitioners, but also serves to synthesize various theoretical perspectives on, for example, strategic fit, transition timing, and performance management [64].

As such, the CAMO format allows for substantive (or contextualized) theory development in terms of relationships between context, outcome patterns, mechanisms and agency. However, it also enables more formal theory development, especially when studies across different (e.g., industrial, cultural or institutional) contexts serve to decontextualize theory in agency–mechanism–outcome relationships [47,49,52,62,67].

*CAMO principles enable practitioners to apply research evidence.* DS-based studies also demonstrate that a synthesis of the extant body of research evidence in CAMO format (e.g., design principles) can effectively inform practitioners in addressing major problems and challenges [9,12,48,50,51,61,68]. For example, Bevan and coauthors [50] formed a team of organization development (OD) practitioners, researchers and healthcare poli-

cymakers to successfully drive organizational change in the National Health Service in the UK. They developed a set of design principles depicting "10 high impact changes" that enabled senior executives to apply radical improvement ideas to their strategies and goals; these principles provided them with the necessary evidence and reassurance to "take the plunge" [50] (p. 147). As such, their evidence-based design principles appear to "increase the probability that a design will be successful, at the same time giving practitioners what they have always sought (and not always received) from OD, namely, broad-based solutions that are said to work" [50] (p. 147).

*Validating serves to grow the body of CAMO evidence & identify research opportunities*. Several studies illustrate how validation efforts help to grow the body of knowledge [48,51,53,55]. For example, the (CAMO-like) design principles for university spinoff creation developed by Van Burg et al. [66] were subsequently replicated in a study of several US-based universities [86] and extended in a study of other venture creation programs on several continents [87]. Moreover, by engaging directly in validation work or by reviewing the evidence basis, one is also likely to identify knowledge gaps and research opportunities in the extant body of knowledge (e.g., [49,62,66,67,88]). For instance, several research opportunities identified by Van Burg et al. [66] informed subsequent studies of transparency, conflicts of interest and fairness in spinoff creation [89–91].

## 5. Connecting Design and Science

The need to forge a seamless connection between "what is" and "what could be" [1] raises fundamental challenges for management scholars: the science mode is retrospective in making sense of the world as it is, and the design mode is prospective in envisioning and creating a future state of the world. In Figure 1, we outline a methodological framework that connects these modes. This framework arises from the review in the previous section and draws on two additional premises: (1) a change initiated by practitioners and/or scholars is meaningful if we can determine whether and why it leads to desired outcomes (i.e., solutions); (2) an understanding of how certain outcomes arise is useful if we can deploy it to initiate change, to make things better.

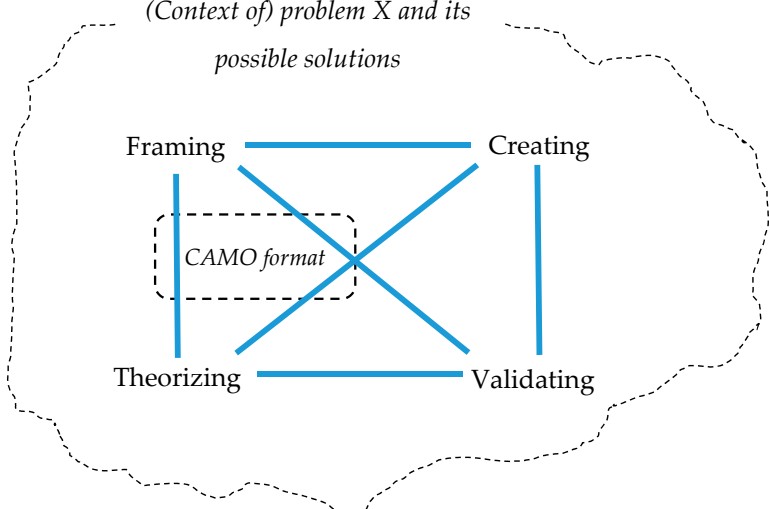

**Figure 1.** Framing, theorizing, creating and validating in design science, including the role of CAMO knowledge.

DS focuses on solving problems, that is, changing extant systems and practices into desired ones. However, a DS approach typically treats the initial situation/problem as ill-defined, which at that stage also blurs the line between problem and context [11,56]. Thus, a key step in any DS project is to frame and theorize about the problem and the problem-solving process [92]. The frame serves as a gateway to creating and validating (the

constructs and models underlying the) solutions. Similar to March and Smith [3], we thus differentiate the high-level category of design into framing and creating and the metaphor of science into theorizing and validating.

In the framework outlined in Figure 1, *framing* refers to the act of exploring which (set of) construct(s) can provide a gateway for discovering and understanding the problem and solution space [39,92,93]. Framing can involve individual cognition as well as the social construction of a particular problem/solution space. The framing notion resonates well with those emphasizing that theory development is an act of creative discovery [32,94] and is pivotal in many DS studies reviewed earlier (e.g., [48,52,53,57,59]).

*Creating* in DS involves the act of conceiving and realizing a new(ly perceived) artifact such as a strategy practice [54], entrepreneurial practice [60,66], intervention strategy [48,55] or managerial tool [10,57,61,68]. Framing and creating activity often go together, with many iterations back and forth. Most work by management practitioners does not go beyond this design space, especially when they create solutions within existing frames or when these frames undergo only minor changes to facilitate decisions on solutions.

The science mode in Figure 1 is further disentangled into theorizing and validating activities. *Theorizing* is about developing key concepts into well-defined constructs and formulating causal propositions and models which are generalizable as well as applicable to individual cases [27]. Most of the DS studies reviewed earlier engage in some form of theorizing (e.g., [12,48,52,54]), although none of these studies focus on theoretical output as such. Figure 1 implies that theorizing not only guides and is affected by validation efforts but also interacts with framing and creating activity.

Finally, *validating* in DS involves the evaluation of (preliminary) frames or artifacts. Venable et al. [4] mapped the various validation methods in terms of two dimensions: ex-ante and ex-post evaluation (i.e., prior to versus after creating the frame/artifact) and naturalistic and artificial evaluation (e.g., field studies versus intervention/pilot studies). Artificial evaluation can be further differentiated in "alpha tests" in which the designer assesses whether and how the artifact performs in the (initial) setting where it was created (e.g., [9,48]), and "beta tests" in which others implement and assess the artifact in other settings (e.g., [48,59]). Overall, methods for artificial validation serve to assess whether and why the created artifact performs as expected—for example, in terms of usability, reliability, fairness and productivity (e.g., [10,50,61,68]). Naturalistic methods are more appropriate for testing theoretical propositions and models—in terms of their internal and external validity, reliability and generalizability (e.g., [12,62,64]).

Notably, the framing, creating, validating and theorizing acts appear to be complementary and feed on each other. For example, to be able to theorize in terms of a causal model, one first must explore what the most appropriate conceptual framing of the problem at hand is [29,57,67]. In their use of theory, scholars can thus act as both scientists and designers; that is, they can aim to develop a theoretical representation of the problem as it is and/or use a conceptual frame to explore potential future solutions.

Figure 1 also serves to demonstrate the potential role of CAMO-formatted knowledge in connecting the four DS activities. CAMO propositions, also known as design rules, are especially instrumental in connecting the framing and theorizing acts but can also help connect the (emerging) body of knowledge that arises from framing and theorizing to creating and validating solutions (or other artifacts) (e.g., [12,59,60,64,67]). Notably, the framework in Figure 1 does not explicitly refer to "what" is framed, created, theorized and validated. In this respect, DS researchers tend to embrace a broad set of potential *artifacts*, such as management practices, tools, constructs, models, (research) methods, conceptual frameworks, and design principles [33,55,59,61,63,67,68]. While many of these artifacts can be the output of any of the four activities in Figure 1 and be used as input in subsequent activities, the DS studies reviewed in Section 4 tend to pay much more attention to the development of new practices and tools than to theoretical constructs and models (e.g., [9,51,56,61,68]), probably due to the problem-solving orientation of DS work [6,43].

The research cycle in Figure 1 appears to have no formal starting point, so one can start anywhere. For example, design-oriented practitioners often start with generating "workable knowledge solutions without having a fully formed theoretical understanding of the organizational components or systems they are designing" [46] (p. 436). That is, the problem is first framed, which informs the creation of initial (ideas for) solutions; these solutions are then validated, for example, by pilot-testing them in one or two sites within the organization, which fuels efforts to theorize about the underlying causal relations (e.g., [48,59]). Alternatively, DS work can also start with theorizing, for example, by reviewing the literature on a particular (theoretical or practical) problem, including an assessment of the validation (evidence base) of various models and practices identified in the literature; subsequently, they frame a direction for problem-solving and create a solution prototype that then needs to be tested (validated), which typically invites a number of iterations in the entire research cycle (e.g., [10,55,68]). Other work informed by DS may cover only parts of the cycle, leaving the remaining activities to future work; for example, review studies of a particular problem area can focus on synthesizing existing theories and their validation in the form of a set of design propositions (e.g., [47,62]) that inform design and intervention efforts in subsequent studies.

## 6. Concluding Remarks

Kurt Lewin's phrase "There's nothing so practical as good theory" [95] (p. 169) eloquently expresses the dual role of theory in providing conceptual understanding as well as guiding action. The DS perspective outlined in Figure 1 suggests these two roles—theorizing and framing—operate in a cycle, keeping one another in check. An action becomes a basis for knowledge if it is expressed in theoretical terms, and a conceptual or theoretical expression is useful if it can help frame a practical problem and inform the development of its solution. Over many years since Lewin's quote, the two roles have increasingly diverged in management research, separating academia and practice and giving rise to major tensions between rigor and relevance [21,23,96]. For most management scholars, the pursuit of theory has become an end in itself—with the main criterion for the good theory being whether the theory can be validated.

The DS framework outlined in Figure 1 provides an inclusive approach to theory and theorizing in the context of understanding and solving real problems. This basic methodological framework incorporates the conception of theorizing as imposing conceptual order on the empirical complexity of organizations and their management [27,28] and an act of creative framing and discovery [31,32]. Moreover, the studies reviewed in Section 4 suggest that the synthetic nature of CAMO-formatted knowledge at the interface between design and science promotes replication and extension efforts and thus knowledge accumulation.

Our literature review also demonstrates that framing can be highly instrumental in challenging a given mainstream theory and thereby (at the level of a broader body of knowledge) reduce the risk that researchers remain constrained to this theory [21,97]. Moreover, we observed that many DS-based studies attempt to connect and integrate descriptive-explanatory and prescriptive-normative knowledge. We codified this insight in the context–agency–mechanism–outcome (CAMO) framework, which can be used to develop a set of propositions/principles that together represent the extant body of knowledge in a particular subfield of management. The CAMO framework can thus facilitate the ongoing dialog between researchers and practitioners to build a "shared memory" [13] that promotes the application as well as articulation and accumulation of knowledge.

Without the need to think of theory as actionable and practical, scholars tend to become less sensitive to the context of theoretical application [40]. This is best exemplified by the *ceteris paribus* qualifier in how theoretical propositions are typically formulated—a shorthand for what Schön [98] termed a retreat to the "high ground" of well-defined problems with a clear solution space. Moving away from the purity of theoretical abstraction, in the face of practical problems that require creative solutions, one was left in the "swampy

lowland" of confusing problems that defy simple solutions [98]. In their attempts to enlist the help of (e.g., management) theories, practitioners often face an incoherency problem: there were many theories of the same thing, often incoherent when viewed collectively [99]. In this situation, the practitioner does not (only) seek a theory *of*, but (also) a theoretical framing *for* the problem, that is, a gateway to its solution. The DS framework developed in this article may thus serve to make the notion of theory whole again, by interlacing retrospective theorizing and prospective framing—in the field of management, but also in other fields such as information systems [4], engineering design [34] and product design [100].

**Author Contributions:** Conceptualization, A.G.L.R. and D.D.; methodology, A.G.L.R.; investigation, A.G.L.R.; writing—original draft preparation, A.G.L.R. and D.D.; writing—review and editing, A.G.L.R. and D.D.; visualization, A.G.L.R. All authors have read and agreed to the published version of the manuscript.

**Funding:** This research received no external funding.

**Institutional Review Board Statement:** Not applicable.

**Informed Consent Statement:** Not applicable.

**Data Availability Statement:** The data presented in this study are available in Section 4 of this article.

**Acknowledgments:** Earlier versions of this paper were presented at the *Entrepreneurship as Design* workshop in Gothenburg (December 2019) and at the Design Theory SIG workshop of the *Design Society* in Paris (January 2020). The authors are grateful for the constructive feedback from many participants in these events.

**Conflicts of Interest:** The authors declare no conflict of interest.

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
