# Peer review of "Mixing Oil with Water: Framing and Theorizing in Management Research Informed by Design Science"

_designs_

Round 1
Reviewer 1 Report
I read this article with great interest. It represents an important contribution to concepts such as design science and framing. It exceptionally well summarises and presents the tradition of design science, the tool to bridge the world of practitioners and the academic world in, for example, management or entrepreneurship research. For both, practitioners and academic researchers it provides an important description, a roadmap, of how to deal with the two opposite poles of the theory of and the theory for in their particular research.
This paper provides a very well-designed and presented analysis, including the use of relevant literature and references.
Author Response
Many thanks for your very positive assessment of our manuscript.
Reviewer 2 Report
When reviewing scientific papers for publication, I usually start with a general overview in terms of a structure, abstract, literature review, methodology, findings of the research, discussion, conclusions, as well as limitations of the study.
In the assessment of the paper submitted for the review, I specifically focused on the discussed issues, applied research methods and the scope of analysis of research results, as well as substantive content of the article and its structure.
I think the writing of this article has a certain level, the only suggestion that Figure 1 can be more clear, it is not very clear at present.
Author Response
Many thanks for your review. Your only point of improvement is the "suggestion that Figure 1 can be more clear, it is not very clear at present." You did not specify what is unclear in the figure, and therefore we rewrote the explanation of the figure in the text. Please note we also adapted Figure 1 in response to a suggestion of reviewer 3, by referring to the context of the problem and its potential solutions.
Reviewer 3 Report
The contribution reviews how DS is used by management researchers.
The contribution sets out to outline how theory and theorizing is underdeveloped in the discourse on combining generative design and explanatory science and applies this to the field of management research.
The aim is to demonstrate the value of theorising and framing, in particular in design-oriented approaches.
The paper is an interesting contribution to DS, and could be of benefit to readers from disciplines other than management-oriented practitioners.
I believe the CAMO format proposes a useful adaptation of the CIMO format. The structure of CAMO is perfectly intelligible and the bibliography exhaustive.
However, the following improvements could support the contribution:
- The distinction of the CAMO from the CIMO format could be supported by adding a table that synthesises the CIMO format (similar to the table in line 418), perhaps summarising the references that are listed in line 288.
- Although the focus of DS is to frame about the problem, perhaps figure 1 could indicate the 'context', as well as the 'problem X', as 'the DS approach blurs the line between problem and context' (line 284), which would support the CAMO format.
- The abstract states that this contribution may be of interest to 'design-oriented disciplines that tend to adopt a rather intuitive (undefined) notion of theory’, however these disciplines are not further defined or listed in the paper. Perhaps a clarification of the disciplines in the concluding remarks, or a description of the disciplines in the abstract, could indicate who could implement the CAMO format in more detail.
Furthermore, the following details are highlighted for revision:
42 remove the bracket between [2-6] and [14,17], or perhaps the formatting could be similar to the references numbering in line 206
206 Perhaps avoiding the full reference list could help with the numbering of references in the text. For example, reference 47 in line 202 is followed by reference 52 in line 218.
255 please explain who the lead designer is in this example.
276, 356, 461 various references in the paper are listed with 'e.g.', perhaps 'e.g.' could be removed.
285 the term 'CMO' is used, which might stand for CIMO. If correct, perhaps CMO could be described.
Author Response
Many thanks for your compliments and positive assessment of our manuscript. Here is a summary of how we've addressed your various suggestions and points of improvement:
1. The distinction of the CAMO from the CIMO format could be supported by adding a table that synthesises the CIMO format (similar to the table in line 418), perhaps summarising the references that are listed in line 288.
RESPONSE: rather than adding another table, we extended the explanation (in lines 330-332) of the difference between CIMO and CAMO. This revised text (330-334) says: "The key difference with CIMO is that the latter leaves the agent dimension somewhat unspecified, by merely describing the intervention/action (and not who the agent is behind this intervention); the main difference between CIMO and CAMO therefore is that the Intervention dimension is replaced by the Agency dimension."
2. Although the focus of DS is to frame about the problem, perhaps figure 1 could indicate the 'context', as well as the 'problem X', as 'the DS approach blurs the line between problem and context' (line 484), which would support the CAMO format.
RESPONSE: Very good suggestion; because the boundaries between problem and context are indeed blurred, we adapted the label of the dashed outer lining in Figure 1 to: (Context of) problem X and its possible solutions.
3. The abstract states that this contribution may be of interest to 'design-oriented disciplines that tend to adopt a rather intuitive (undefined) notion of theory’, however these disciplines are not further defined or listed in the paper. Perhaps a clarification of the disciplines in the concluding remarks, or a description of the disciplines in the abstract, could indicate who could implement the CAMO format in more detail.
RESPONSE: Excellent suggestion! We added three examples of other fields/disciplines in the last sentence of section 6.
42 remove the bracket between [2-6] and [14,17], or perhaps the formatting could be similar to the references numbering in line 206
RESPONSE: we adapted the referencing to these sources accordingly.
206 Perhaps avoiding the full reference list could help with the numbering of references in the text. For example, reference 47 in line 202 is followed by reference 52 in line 218.
RESPONSE: we're not sure what you exactly mean with "avoiding the full reference list"; we assume your suggestion is to avoid the listing of the 28 sources (initially selected) in line 206. We believe that not giving this list here would reduce the transparency of the review approach adopted, so we kept the text as it is.
255 please explain who the lead designer is in this example.
RESPONSE: between brackets, we inserted the name of the lead designer.
276, 356, 461 various references in the paper are listed with 'e.g.', perhaps 'e.g.' could be removed.
RESPONSE: indeed, because it's quite evident to readers that we're only giving examples from the broader literature database, the "e.g." can be removed in these instances; we changed the text accordingly.
285 the term 'CMO' is used, which might stand for CIMO. If correct, perhaps CMO could be described.
RESPONSE: we specified CMO by writing it out in full: Context-Mechanism-Outcome (CMO); notably, it's not the same as CIMO, as explained in the text that follows in 285-289.